# Mechanisms of Calorie Restriction: A Review of Genes Required for the Life-Extending and Tumor-Inhibiting Effects of Calorie Restriction

**DOI:** 10.3390/nu11123068

**Published:** 2019-12-16

**Authors:** Toshimitsu Komatsu, Seongjoon Park, Hiroko Hayashi, Ryoichi Mori, Haruyoshi Yamaza, Isao Shimokawa

**Affiliations:** 1Department of Pathology, Nagasaki University School of Medicine and Graduate School of Biomedical Sciences, Nagasaki 852-8521, Japan; komatsut@nagasaki-u.ac.jp (T.K.); psj1026@nagasaki-u.ac.jp (S.P.); hayashih@nagasaki-u.ac.jp (H.H.); ryoichi@nagasaki-u.ac.jp (R.M.); 2Section of Pediatric Dentistry, Division of Oral Health, Growth and Development, Faculty of Dental Science, Kyushu University, 3-1-1 Maidashi, Higashi-ku, Fukuoka 812-8582, Japan; hyamaza@dent.kyushu-u.ac.jp

**Keywords:** calorie restriction, FoxO transcription factor, sirtuin, neuropeptide Y, pleiotropy of CR genes

## Abstract

This review focuses on mechanisms of calorie restriction (CR), particularly the growth hormone (GH)/insulin-like growth factor-1 (IGF-1) axis as an evolutionary conserved signal that regulates aging and lifespan, underlying the effects of CR in mammals. Topics include (1) the relation of the GH-IGF-1 signal with chronic low-level inflammation as one of the possible causative factors of aging, that is, inflammaging, (2) the isoform specificity of the forkhead box protein O (FoxO) transcription factors in CR-mediated regulation of cancer and lifespan, (3) the role for FoxO1 in the tumor-inhibiting effect of CR, (4) pleiotropic roles for FoxO1 in the regulation of disorders, and (5) sirtuin (Sirt) as a molecule upstream of FoxO. From the evolutionary view, the necessity of neuropeptide Y (Npy) for the effects of CR and the pleiotropic roles for Npy in life stages are also emphasized. Genes for mediating the effects of CR and regulating aging are context-dependent, particularly depending on nutritional states.

## 1. Introduction

Laboratory rodents often continue to gain body weight after puberty until certain age points under standard husbandry conditions in which animals can freely access food, often called ad libitum (AL) feeding. In contrast, restriction of food intake in mice by 30% to 40%, when initiated at young age, e.g., 12 weeks of age, limits the gain in body weight [1]. Dietary regimens involving food restriction, here referred to as calorie restriction (CR), reduce morbidity and mortality in experimental animals compared with AL feeding animals [1]. Since the original report on the effects of CR in 1935 [2], many laboratories have confirmed the effects of CR and investigated the underlying molecular mechanisms. Notably, a recent meta-analysis of lifespan studies in laboratory rodents described various responses to CR, that is, no extension or even shortened lifespans in CR rodents [3]. The analysis also suggested potentially substantial effects of the genotypes of animals as well as husbandry on experimental outcomes in CR studies. A genome-scale metabolic model and transcriptome data in the male Sprague Dawley rat liver also revealed varied metabolic responses of the liver with different levels and durations of CR [4]. Recent studies have also identified sexual dimorphism in physiological responses to CR [5,6]. Kane et al. [6] found that female rodents may have a greater response to CR than male rodents, with modulated sensitivities to mechanistic target of rapamycin (Mtor), growth hormone (GH), insulin-like growth factor-1 (IGF-1) or fibroblastic growth factor 21 and greater inflammatory and mitochondrial health responses [6]. Therefore, the effects of CR may not be universal as originally expected. Nonetheless, CR models in a range of organisms have contributed to our understanding of the aging process.

Epistasis analyses using mutant strains in lower organisms such as *Caenorhabditis elegans* (*C. elegans*) have revealed genes required for the effects of CR (referred to here as CR genes) and the signal pathways mediating the effects of CR. In *C. elegans,* a number of genes such as *aak-2*, *daf-16*, *skin-1*, *clk-1*, and *pha-4* have been reported to be associated with the life-prolonging effect of CR [7]. Some of these genes also mediate the effects of CR in mice. Previous studies also reported that mutations of single genes (referred to here as longevity genes) can extend lifespan even in AL feeding animals. Many of these genes can be functionally categorized into genes associated with nutrient sensing or metabolic responses [8]. Among these gene mutations, reduction- or loss-of-function mutations of genes in the growth hormone (GH)-insulin-like growth factor-1 (IGF-1) signaling consistently extend lifespan in a range of organisms [8]. Since CR is known to decrease the plasma concentration of GH and IGF-1, the GH-IGF-1 pathway is considered an evolutionary conserved pathway for longevity and a main aspect of the mechanism of CR [9].

Thus far, a total of 112 CR genes in yeast, 62 in nematode, 27 in drosophila, and seven in mice have been identified and are listed in the database [10]. Among these genes, forkhead box protein O 3 (*Foxo3)* and sirtuin 1 (*Sirt1*) genes are common in mice, nematodes, and flies. CR and longevity gene models have elucidated signal pathways for the extension of lifespan, although the signal pathways are context dependent. This review will focus on FoxO transcription factors and the sirtuin deacetylases that are upstream of FoxO. We also discuss roles for neuropeptide Y *(Npy),* which is essential for the effects of CR in mice and thus mammals.

## 2. A Central Role for GH and IGF-1 in the Regulation of Lifespan

Previous studies reported that a single gene mutation can prolong the lifespan of experimental animals under AL conditions of standard diets. Genetic analyses of long-lived strains of nematodes identified a mutation in a single gene, *age-1* [11]. This gene was later found to encode a component of phosphoinositide 3-kinase (PI3K) that is important for growth factor signals such as those modulated by IGF-1 [12]. Kenyon et al. showed that mutation of *daf-2*, which is associated with resistance larval development, prolongs the lifespan of *C. elegans*, and *daf-16* is required [13]. Daf-2 and Daf-16 correspond to the receptor for IGF-1 and the FoxO transcription factor in mammals, respectively. Thus, in *C. elegans*, attenuated IGF-1 signaling promotes activation of Daf-16, leading to transcriptional modulation of its target genes and extended lifespan. Even in *Drosophila*, mutations in genes in insulin-like signaling such as *Inr* (insulin-like receptor) and *chico* (insulin-receptor substrate) can extend the lifespan [14]. In these conditions, *dFoxo* is also required.

The lifespan of *C. elegans* can also be extended by suppression of target of rapamycin (Tor in mammalians, mechanistic target of rapamycin, Mtor), which is downstream of IGF-1 signaling and promotes cell proliferation and division when nutrients are abundant [15]. Mtor forms complexes (mTORC1 and mTORC2) with other molecules in nutrition and energy rich conditions. These complexes activate transcription and translation when insulin and growth factors concomitantly rise. Mtor complexes promote protein synthesis and cell division while inhibiting autophagy. All genetic manipulations that suppress Mtor identified thus far can extend the lifespan of *Drosophila* [16].

In mammals, GH is upstream of the IGF-1 signal. GH is competitively controlled by Somatostatin and GH-releasing hormone (Ghrh), which are secreted from hypothalamic neurons. In mice, reduction-of-function gene mutations in molecules involved in the signal between Ghrh and IGF-1 consistently prolong lifespan [17]. Furthermore, longevity is achieved by inhibition of mTORC1 by rapamycin [18], deletion of the *S6K* gene [19], and suppression of Mtor [15,20].

Together these results in a range of experimental animals indicate that signal attenuation of the IGF-1 signal, activation of FoxO transcription factor, and suppression of Mtor are key mechanisms for slowing aging and prolonging lifespan (Figure 1). However, it should be noted that the life-extending effect of the reduced IGF-1 signaling could be sexually dimorphic. In Igf1 receptor (*Igf1r*) gene heterozygotic knockout (*Igf1*^+/−^) mouse models of different genetic backgrounds (129/SvPas and C57BL/6J) [21,22], only female *Igf1r*^+/−^ mice significantly outlived controls, whereas this was not observed for males. Genetic studies in humans also indicate that genetic variations causing reduced IGF-1 signaling are beneficial for survival in old age only in females but not in males [23,24,25]. The mechanisms underlying the sexual dimorphism for longevity in the context of reduction of IGF-1 signaling remain to be explored.

## 3. Possible Relation of the GH-IGF-1 Axis with Inflammaging

Generalized chronic low-level inflammation may be a causative factor for aging and related diseases. This low-level inflammation associated with the aging process is called inflammaging [26]. It is also called sterile inflammation because it is not caused by infection.

Senescent cells that increase in tissues with age not only stop undergoing cell division but also express inflammatory cytokines such as IL-1α, IFNβ, chemokines, matrix metalloproteases, and growth factors, inducing local inflammation in tissues. This phenomenon is called the senescence-accelerated secretory phenotype (SASP) [27]. A recent report showed that the average lifespan of mice can be extended by selectively removing senescent cells expressing p16 ^Ink4a^ from tissues [28]. The removal of senescent cells also suppresses age-related lipodystrophy, tumorigenesis, glomerulosclerosis of the kidney, and so on. This report indicates that senescent cells are involved in the progression of aging in animals.

SASP was reported to be caused by induction of type I interferon (IFN-I) signal by activating transcription of the retrotransposable element long-interspersed element-1 (L1) [29]. In normally proliferating cells, L1 is suppressed by a cell monitoring mechanism, e.g., three prime repair exonuclease 1 (Trex1) and the RB transcriptional corepressor 1 (Rb1). In senescent cells, these repression mechanisms are disrupted, leading to activation of L1 via activation of FoxA1 [29].

During exposure of cells to harmful stimuli and conditions, such as chemicals and ischemia, the injured cells excrete or secrete various molecules such as ATP, uric acid, oxidatively modified DNA, and aggregated proteins, which are called damage-associated molecular pattern molecules (DAMPs) [30]. DAMPs bind to receptors, such as Toll-like receptors, in the cell membrane and cytoplasm of resident macrophages and elicit innate immune and inflammatory responses. The sensor of innate immunity, called the nucleotide-binding domain, leucine-rich-containing family, pyrin domain-containing-3 (NLRP3) inflammasome, is activated, and macrophages secrete inflammatory cytokines and exacerbate inflammatory responses. These inflammatory signals have been reported to damage surrounding parenchymal cells. As with senescent cells, this minor but chronic inflammatory response at the cellular level is thought to be the main factor driving aging and disease, including cancer.

GH is not only secreted from anterior pituitary cells but is also expressed in various cells including immune cells. The GH receptor (Ghr) is also expressed in a wide range of cells, and the GH signaling system is activated by endocrine action as well as by autocrine and paracrine actions. Recent studies suggested that the sensitivity of the GH-IGF-1 signal increases with age and that DAMP inflammasome is likely to be activated [31]. This activation of the inflammasome is suppressed in long-lived Ghr-deficient mice [31]. Since CR also attenuates GH-IGF-1 signals, suppression of the inflammasome could be a main component of the anti-aging mechanism of CR.

## 4. Differential Regulation of Cancer and Lifespan by CR via FoxO Transcription Factors

Although multiple studies have used a variety of CR regimens in invertebrates and vertebrates, most of the CR regimens extend lifespans in these organisms. To analyze the signal pathways mediating the life-prolonging effects of CR, epistasis experiments in which mutation of a single gene causes attenuation or abrogation of the life-prolonging effect of CR have been conducted, mostly in *C. elegans*. These experiments have identified CR genes such as *daf-16*, *skn-1*, *pha-4*, *clk-1*, *aak-2*, and *hsf-1*, although the genes required for the effect of CR may differ depending on methods of CR [7]. These CR genes suggest the importance of regulation of insulin-like signaling, stress response, and mitochondrial bioenergetics.

CR is known to lower plasma concentrations of insulin and IGF-1 in mammals [9]. Therefore, IGF-1 signaling and thus FoxO transcription factors, the mammalian orthologs of Daf-16, were considered to play roles in the effects of CR. As mentioned, loss of Daf-16 abrogates the life-extending effects of CR in *C. elegans* [7]. In mammals, the FoxO transcription factor family includes four isoforms, FoxO1, FoxO3, FoxO4, and FoxO6 [32]. We tested the hypothesis that FoxOs are involved in the effects of CR by two lifespan studies using *Foxo1* and *Foxo3* knockout mice.

For the FoxO1 study, we used *Foxo1*
^+ /−^ mice, because *Foxo1*^−/−^ mice died around embryonic day 11 due to defects in the branchial arches and remarkably impaired vascular development of embryos and yolk sacs, indicating the necessity of *Foxo1* in development of the cardiovascular system [33]. In comparison, *Foxo1*^+/−^ mice grow normally through pre- and post-natal stages. In *Foxo1*^+/−^ mice, expression levels of *Foxo1* mRNA in the examined tissues were reduced by 50% compared with levels in wild-type (WT) mice [34]. Results of the lifespan study indicated that CR extended lifespan in *Foxo1*^+/−^ mice to the same extent as in WT mice; however, unlike CR in WT mice, CR did not significantly reduce the proportion of *Foxo1*^+/−^ mice bearing spontaneously occurring tumors. By contrast, in *Foxo3*^+/−^ and *Foxo3*^−/−^ mice, CR did not extend lifespan [35], indicating the requirement of the *Foxo3* gene in the life-extending effect of CR in mice. With AL feeding, *Foxo3*^+/−^ mice displayed no reduction of lifespan compared with WT mice, while *Foxo3^−/−^* mice showed a slight reduction of lifespan [35]. Although the effect of CR in *Foxo1*-null mice should be investigated, our lifespan studies suggest a differential regulation of cancer and lifespan by CR via *Foxo1* and *Foxo3* (Figure 1).

In early studies using mice with germline-deleted *Foxo1*, -*3*, and -*4* genes, deletion of a single gene had little effect on longevity or cancer development in postnatal life under AL conditions [36]. One exception was female *Foxo3*^−/−^ mice, in which occurrence of pituitary adenoma was accelerated compared with WT and *Foxo3*^+/−^ mice. Conditional, widespread somatic *Foxo* deletion in adult tissues was also achieved using an interferon-inducible Mx-Cre transgene to circumvent embryonic lethality of FoxO1 deficiency [36]. When the three *Foxo* genes were deleted in mice at the same time, tumor incidences were elevated, while lifespan was shortened. In particular, the incidences of thymic lymphoma and hemangioma were increased, compared with groups of mice retaining at least one *Foxo* allele [36]. *Foxo1*, -*3*, and -*4* gene-deleted mice showed premature death due to the tumors. These experiments indicated that FoxO transcription factor genes are complementary as for development of cancers [36]. In other words, even if one *Foxo* gene is deleted, the other *Foxo* genes can substitute in AL conditions. However, our lifespan studies [34,35] indicate differential roles for FoxO1 and FoxO3 in the effects of CR.

The Daf-16 isoform-specific extension of lifespan in the context of reduction of IGF signaling has been also reported in *C. elegans*. The *daf-16* genomic locus encodes three groups of transcripts (a, b, d/f/h) that are transcribed from distinct promoters [37,38,39]. Depending on the experimental setting, Daf-16a and/or Daf-16d/f/h, but not Daf-16b, play a major role in the extension of lifespan [39,40].

Numerous human genetic studies have indicated a correlation of the minor alleles of the *FOXO3* gene with longevity [41]. However, there is no significant correlation of *FOXO1* genotypes with longevity. A genome-wide meta-analysis of 30844 adults of European ancestry from 21 studies confirmed that the known longevity-associated *FOXO3* variant rs2153960 is a genome-wide significant SNP for lowering IGF-1 concentrations [42].

The isoform specificity of FoxO and Daf-16 in the life-extending effect of CR as well as the human genomic studies for longevity suggest that the mechanism of lifespan control has been evolutionarily conserved. A recent study showed that FoxO6 functions in age-related insulin resistance and inflammation in rats with AL feeding, although CR was not investigated [43]. Thus, potential roles for FoxO4 and FoxO6 in the effects of CR remain to be elucidated.

How FoxO3 regulates lifespan remains unclear. Our microarray analysis in *Foxo1*^+/−^ and *Foxo3*^+/−^ CR mouse liver show that many genes are differentially regulated in these mice under conditions of CR (Appendix A). The pathway analysis of the differentially expressed genes between WT-CR and *Foxo1*^+/−^ CR mice as well as WT-CR and *Foxo3*^+/−^ CR mice also suggests that a number of inflammation and immune response pathways are activated in *Foxo3*^+/−^ CR mice. By contrast, in *Foxo1*^+/−^ CR mice, some T-cell functions may be down-regulated. FoxO3 is reported to prevent excess activation of interferone (IFN)-I in response to viral infection [44]. A ternary complex consisting of FoxO3, nuclear co-repressor 2 (Ncor2) and histone deacetylase 3 (Hdac3) exists on the promoter of interferon regulatory factor 7 (*Irf7*); a loss of FoxO3 enhances histone acetylation in the promoter of *Irs7* gene, resulting in increased levels of *Irf7* mRNA in macrophages [44]. An experimental stimulation of IFN-I activated the PI3K/Akt pathway, which in turn led to FoxO3 degradation. FoxO3 could play a role in optimizing host defense mechanisms against harmful stimuli as well as under unstimulated conditions by limiting the transcription of Irf7 and then Irf7-induced target inflammatory genes. This process could minimize activation of the inflammasome. Reduction of FoxO3 may disrupt the FoxO3-Irf7 regulatory circuit under CR conditions.

## 5. FoxO1 Mediates the Tumor-Inhibiting Effect of CR

A tumor-inhibiting but not life-extending effect of CR has been reported in *Nrf2*^−/−^ mice [45]. The Nrf2 transcription factor belongs to the cap’n’collar basic-leucine zipper (CNC-bZIP) family. Under unstressed cellular conditions, Nrf2 binds to Keap1 in the cytoplasm and is degraded by the ubiquitin-proteasome system [46]. When the redox environment changes due to oxidative stress, Nrf2 and Keap1 dissociate and Nrf2 moves into the nucleus. Nrf2 forms heterodimers with other transcription factor cofactors such as Maf, binds to the antioxidant response element (ARE) in the promoter of target genes, and promotes the gene expressions of antioxidants and phase II detoxification enzymes.

The role of Nrf2 in the tumor-inhibiting effect of CR was tested in a two-stage skin carcinogenesis model using 7,12-dimethyl-benz (a) anthracene (DMBA) and 12-O-tetradecanoyl phorbol 13-acetate (TPA) [45]. In WT mice, CR significantly suppressed the development of skin tumors; in *Nrf2*^−/−^ mice, the CR-mediated inhibitory effect on tumor growth was absent. In *Nrf2*^+/−^ mice, the inhibitory effect was attenuated, suggesting a gene-dose dependency of *Nrf2* in the inhibition of tumor formation by CR. However, even in *Nrf2*^−/−^ mice, the life-extending effect of CR was preserved [45]. These results suggest that Nrf2 plays a major role in the tumor-inhibiting effect of CR but not for lifespan extension.

We also conducted a two-stage skin carcinogenesis model experiment to confirm the potential role for FoxO1 in the effect of CR, following the protocol of Pearson et al. [46] Appendix B. Briefly, DMBA was applied in the dorsal skin of mice at 31 weeks of age; after two weeks, TPA treatment was initiated. Skin papilloma started occurring six weeks after DMBA application, and thus four weeks later starting TPA treatment (Figure 2A). Our findings showed that haploinsufficiency of *Foxo1* accelerated the occurrence of skin papilloma, particularly in CR conditions (WT-CR vs. *Foxo1*^+/−^ CR, *p* = 0.0103 by log-rank test; Figure 2A), confirming the role for FoxO1 in the tumor-inhibiting effect of CR, as in Nrf2.

TPA promotes epidermal hyperplasia in the pre-neoplastic phase via reactive oxygen species (ROS) production and inflammation [47]. We assessed the epidermal thickness during TPA treatment and the proliferative activity of epidermal cells after TPA treatment. Epidermal thickness was greater in *Foxo1*^+/−^ mice than in WT mice (genotype, *p* = 0.0331 by 3-f ANOVA: Figure 2B), although diet did not affect the thickness (diet, *p* = 0.2993 by 3-f ANOVA). Epidermal cell proliferation was greater in *Foxo1*^+/−^ mice (genotype, *p* = 0.0193 by 3-f ANOVA; Figure 2C); the rate of apoptosis did not differ between WT and *Foxo1*^+/−^ mice (data not shown). Thus, the reduced FoxO1 promotes epidermal cell proliferation in response to TPA.

We also evaluated macrophage infiltration in the dermis as an index of inflammation. Although the number of macrophages was less in the CR groups compared with the AL groups, indicating an effect of CR (diet, *p* = 0.0001 by 3-f ANOVA: Figure 2D), the number of macrophages did not significantly differ between WT and *Foxo1^+/−^* mice (genotype, *p* = 0.1374 by 3-f ANOVA).

We further analyzed the expression of genes associated with inflammatory stimuli. TPA induces cyclooxygenase-2 (COX-2), an enzyme that catalyzes arachidonic acid to prostaglandin G2, i.e., produces chemical mediators of inflammation, in the mouse skin [48]. The mRNA expression levels of COX-2 were elevated after TPA treatment (Figure 2E). The expression levels were significantly lower in WT-DR mice compared than other groups of mice; the effect is diminished in *Foxo1*^+/−^ CR mice (diet; *p* = 0.0123; genotype x diet; *p* = 0.0399 by 3f-ANOVA). Although other inflammatory responses induced by TPA were mostly reduced in response to CR, even in *Foxo1*^+/−^ CR mouse skin (data not shown), reduction of FoxO1 in the CR condition might affect the regulation of prostaglandins formation and then impair the tumor-inhibiting effect of CR.

The above results indicate that both FoxO1 and Nrf2 are necessary for the tumor-inhibiting effect of CR. Some Nrf2 target genes are also regulated by FoxO1 [49], such as heme oxygenasae (decycling) 1 (Hmox-1), glutamate-cysteine ligase, and catalytic subunit (Gclc) genes. Gclc is the rate-limiting enzyme in the synthesis of Glutathione (GSH). We also analyzed the expression of Nrf2-target genes and found that *Gclc* and *Hmox-1* mRNA expression levels were lower in *Foxo1^+/−^* mice than in WT mice (only the data of *Gclc* mRNA are shown in Figure 2F; genotype; *p* = 0.0207: diet; *p* = 0.0175: genotype x diet; *p* = 0.5809 by 3f-ANOVA). In a study in *C. elegans*, Daf-16 and Skn-1 target genes were also reported to be preferentially regulated to induce conditions of mitohormesis [49], which could be a mechanism of CR.

## 6. Pleiotropic Roles for Foxo1 in Disorders

As indicated above, reduction of FoxO1 diminishes the anti-tumor effect of CR. However, reduction of FoxO1 does not affect lifespans in AL and CR conditions [34]. Multiple studies have reported beneficial effects of reduced FoxO1 on lesions or disorders, although these results depended on nutritional states. In AL conditions, *Foxo1*^+/−^ mice showed accelerated skin wound healing with enhanced keratinocyte migration, reduced granulation tissue formation, and decreased collagen density, accompanied by an attenuated inflammatory response, compared with WT mice [50]. *Foxo3*^−/−^ mice did not show any significant phenotypes for skin wound healing and inflammation. Reduction of FoxO1 also ameliorates glucose intolerance and insulin insensitivity in diabetic models [51], probably due to diminution of gluconeogenesis. Indeed, FoxO1 is reported to increase hepatic gluconeogenesis through upregulation of glucose 6-phosphatase (G6pc) and phosphoenolpyruvate carboxykinase (Pck1) in CR and fasting conditions [42].

Stress resistance is one of the hallmarks in CR animals, although whether this trait directly causes the extension of lifespan remains unclear. We tested stress resistance in *Foxo1*^+/−^ mice using the endotoxin shock model following the procedures of Kamohara [52]. The survival rates and endoplasmic reticulum (ER) stress were monitored after lipopolysaccharide (LPS) injection. ER stress-responsive alkaline phosphatase (ESTRAP) mice [53] were used to monitor ER stress in vivo (details are described elsewhere [52]). In ESTRAP mice, secreted alkaline phosphatase (SEAP) is constitutively expressed, and SEAP activity in the blood is reduced by ER stress [53]. The experiment was performed using mice at six months of age. The results showed that the survival rate was extended by CR, particularly in *Foxo1*^+/−^ mice (diet, *p* = 0.0263; genotype × diet, *p* = 0.0498 by likelihood ratio test: Figure 3A). After LPS injection, SEAP activity decreased until 8 h in each group and then stayed constant or recovered between 8 and 24 h (Figure 3B). Between 2 and 8 h after LPS injection, that is, in the acute phase of ER stress, CR attenuated ER stress (diet, *p* < 0.0001 by 3-f ANOVA), and the attenuating effect was greater in *Foxo1*^+/−^ mice compared with *Ctrl* mice (genotype, *p* = 0.0093 by 3-f ANOVA: Figure 3B). These data indicate that reduction of FoxO1 enhances stress resistance in both AL and CR conditions. Therefore, in the present experimental setting, reduction of FoxO1 could enhance stress resistance, particularly in the CR condition.

In summary, these findings demonstrate that FoxO1 exerts pleiotropic effects on disorders depending on nutritional conditions.

## 7. Sirtuin as a Molecule Upstream of FoxO

Sirtuins (Sirts) are evolutionarily conserved proteins that catalyze the deacetylation and adenosine diphosphate (ADP) ribosylation of target proteins using the oxidized from of nicotinamide adenine dinucleotide (NAD^+^) as a coenzyme [54]. Since overexpression of *Sir2p*, one of the sirtuin genes in budding yeast, was reported to extend replicative lifespan of yeast, many studies have been conducted to demonstrate the importance of Sirts in the regulation of aging in various experimental models. Although findings in the early phase of investigation were controversial, re-examinations have confirmed Sirts as key molecules in the regulation of aging [55].

Although results in epistasis analyses using mutant strains in lower organisms are sometime affected by experimental settings such as CR or CR-like regimens [7,55], a number of studies have indicated the necessity of *Sirt1* in the life-extending effect of CR. In *Drosophila*, deletion of *Sirt1* (*dSir2*) in the fat body abolishes the life-extending effect of CR [56].

There are seven mammalian orthologs of nematode Sir2 (Sirt1–7) [54]. Sirt1, Sirt6, and Sirt7 exist mainly in the nucleus; Sirt2 is localized in the cytoplasm and Sirt3, Sirt4, and Sirt5 are in mitochondria [54]. Sirt targets in the nucleus include histones, transcription factors, and transcriptional regulators, which control gene expression by deacetylation. One example is the deacetylation of FoxO1 and PGC-1α by Sirt1 in the liver and increased expression of target genes related to gluconeogenesis and fatty acid oxidation [54].

Sirt in mitochondria regulates mitochondrial function, that is, energy metabolism, by deacetylating mitochondrial constituent proteins, energy metabolic pathway proteins, and mitochondrial transcription factors. Previous studies have shown that Sirt3 is deeply involved in the deacetylation of mitochondrial proteins [57].

SIRT4 and SIRT6 mainly function as ADP-ribosylating enzymes [44]. Accumulated evidence has indicated the importance of Sirts in the regulation of energy metabolism and genome instability.

Several studies have indicated the necessity of Sirt1 for the effects of CR in mice. Aged mouse kidney displays pathology such as glomerular and interstitial fibrosis, which is improved by CR [58]. However, this effect of CR is abrogated in *Sirt1*^+/−^ mice. From this model, Kume et al. reported Sirt1-dependent deacetylation of FoxO3 as a key mechanism that promotes mitochondrial autophagy and then reduces oxidative insult in kidneys. Finally, CR extends lifespan in *Sirt1*^+/−^ mice but not in *Sirt1*^−/−^ mice [59]. These findings indicate that one of the mechanisms underlying the effect of CR involves epigenetic regulation of FoxO3 by Sirt1 (Figure 1).

## 8. Roles for Npy in the Effect of CR

An evolutionary perspective predicted that the effect of CR is derived from adaptive responses in animals to harsh conditions such as a famine [60]. Animals should have acquired a system(s) to survive in famine, which frequently occurs in the wild. If not, a species could undergo extinction. When food resources are abundant, animals consume as much food to promote growth and reproduction, while simultaneously storing the remaining energy as fat in the body. In starvation, animals use the stored fat as an energy source by activation of the lipolytic pathway. During food shortage, physiological functions such as reproduction, growth, and heat are suppressed to inhibit excessive consumption of energy. The transition to Dauer states in nematodes and hibernation in some mammals could be extreme examples. The effects of CR may derive from these adaptive processes.

A previous study in *C. elegans* showed that two ciliated neurons that are part of the amphid sensilla (ASI neurons) in the head, in which chemosensitive receptors are present, are necessary for the lifespan extension by CR [61]. These neurons express *skn-1* as described above and are activated when dietary energy is reduced to promote the development of Dauer via the endocrine system [62]. The evolutionary view of CR and the finding in *C. elegans* suggest a possible implication of hypothalamic neurons in the effect of CR in vertebrates.

We focused on Npy, an orexigenic neuropeptide expressed in neurons in the arcuate nuclei of the hypothalamus and the sympathetic nervous system (SNS). Published data indicate that Npy plays a role in neuroendocrine adaptation to negative energy balance in mammals by suppressing activities of growth, reproduction, and heat, whereas it stimulates the glucocorticoid system [63,64]. Npy-overexpressing rats show improvements of cardiac function by regulation of the SNS and marginal extension of lifespan [65]. Npy is also known as a neuroprotective peptide [66]. These traits induced by Npy are consistent with the hallmarks of CR animals. Our previous study also confirmed that CR elevated mRNA levels of Npy in the arcuate nuclei in rats [67].

We tested a potential role for Npy in the effects of CR using *Npy*^−/−^ mice. The life-extending effect in *Npy*^−/−^ mice was attenuated to about one-third compared with WT mice [68]. Oxidative stress tolerance and tumor suppressive effects were also diminished. Initially, Npy deficiency was predicted to compromise neuroendocrine adaptative processes to CR, especially the inhibition of the GH-IGF-1 system, because experiments have shown that Npy is required to suppress Ghrh in fasting [69]. However, hypothalamic *Ghrh* mRNA, plasma IGF-1, insulin, and downstream Mtor in tissues, which could be related to the anti-aging effect of CR, were inhibited as in the WT mice [68]. In addition, plasma concentrations of leptin, adiponectin, and corticosterone in *Npy*^−/−^ mice were also altered similarly as in WT mice [68]. These results indicate the complementary action of the neuroendocrine system in response to CR even without Npy. These findings also suggest that the Npy-related anti-aging mechanism of CR does not overlap with previously noted IGF-1, Mtor, adiponectin, and corticosteroids [68].

*Npy*^−/−^ mice showed a slight increase of food intake normalized body weight in both AL and CR conditions during the lifespan study, indicating a reduction of food efficiency (a unit amount of food or calorie to gain or maintain body weight) compared with WT mice [59], suggesting an increment of energy expenditure in *Npy*^−/−^ mice. In male mice, Npy deficiency increased mortality in CR condition until middle age, probably due to an excess loss of body fat [68]. Indeed, in male *Npy*^−/−^ CR mice, lipolysis and/or thermogenesis was elevated through significant activation of the adrenergic receptor *β3* and hormone-sensitive lipase pathway [70]. The premature death in CR mice was inhibited by administration of acipimox, a lipolysis inhibitor [71]. These findings suggest that the life-extending effect of CR requires inhibition of an excess loss of body energy through antagonizing the SNS. The study also emphasized the substantial role for Npy in the SNS in maintaining fat tissue under CR conditions in male mice. The extent of lifespan extension by CR negatively correlates with rates of white adipose tissue reduction by CR [72], i.e., mouse strains with less reduction of body fat by CR live longer than other strains in which fat is reduced greatly in response to CR. Therefore, the ability of energy preservation by Npy is essential for the life-extending effect of CR.

By contrast, deficiency of Npy may act beneficially in the life-stage under AL conditions. In female *Npy*^−/−^mice with AL feeding, the aging-related increase in body fat was minimized, leading to an improvement of insulin sensitivity via inhibition of inflammation in the fat [70]. Antagonism of Npy may be a promising target for drug discovery to prevent obesity in middle age, although Npy deficiency may cause unintentional weight loss in aged people.

Together these studies indicate that Npy exerts pleiotropic effects in health conditions depending on nutritional conditions.

## 9. Role for Npy in the Tumor-Inhibiting Effect of CR

Npy is expressed abundantly in the brain, adrenal gland, and adipose tissue. Npy circulates in the blood and is secreted locally from the SNS. Minor et al. reported that the tumor-inhibiting effect of CR was diminished in *Npy*^−/−^ mice using a chemically induced skin tumor model [73]. The authors also suggested a role for hypothalamic arcuate nuclei (ARC) in the tumor-inhibiting effect of CR in a model of monosodium glutamate (MSG)-injected mice. MSG induces neuronal cell death in the ARC when injected subcutaneously on postnatal day five. In MSG-injected mice, no Npy-mRNA expression was observed in the ARC but serum Npy was detectable, suggesting a major role for hypothalamic Npy in the tumor-inhibiting effect of CR [73].

Our lifespan study also demonstrated a diminution of the tumor-inhibiting effect of CR in *Npy*^−/−^ mice, suggesting a role for Npy in the tumor-inhibiting effect of CR [68]. The SNS not only releases catecholamines such as norepinephrine (NE) but also secretes Npy [74]. Recent studies have indicated that activation of the SNS exacerbates steatohepatitis, which is now recognized as one of the predisposing conditions of hepatocellular carcinoma (HCC) in obese people [75]. An experimental study also suggested that the SNS promotes HCC via activation of hepatic stellate cells and Kupffer cells [76,77]. Therefore, we investigated a potential role for Npy in a carcinogen-induced HCC model in different nutritional settings including CR as well as high calorie-intake regimens [78].

The complete details of the experiment are described elsewhere [78]. Briefly, HCC was induced by an intraperitoneal injection of diethylnitrosamine (DEN) in male C57BL6/J mice null for *Npy* (*Npy^−/−^*) and their haplotype (*Npy^+/−^*) as a control (Ctrl) on postnatal day 15 [78]. *Npy*^+/−^ mice showed increased lifespan and reduced incidence of spontaneously occurring tumors in response to CR to the same extent as WT mice, indicating that one allele of the *Npy* gene is enough for the effects of CR [69]. Experimental mice, fed a standard diet AL after weaning, were subjected to three dietary regimens at 12 weeks of age and thereafter, that is, standard diet fed AL, 30% CR with the standard diet, and a high fat diet (HFD) fed AL. Mice were killed at 28 and 48 weeks to analyze the occurrence of HCC and growth of HCC in the liver.

CR inhibited the occurrence of microscopic HCC at 28 weeks in *Npy*^−/−^ mice as well as *Ctrl* mice, compared to the AL and HFD groups (diet, *p* = 0.0195 by logistic regression; CR vs. AL, *p* = 0.0192; CR vs. HFD, *p* = 0.0310: Figure 4A), although the baseline proportion of mice bearing microscopic HCC at 28 weeks was slightly but significantly greater in *Npy*^−/−^ mice (genotype, *p* = 0.0426 by logistic regression). Since DEN was administered at 15 days of age before initiating CR, Npy might exert an inhibitory effect on initiation of the carcinogen. At 48 weeks, many mice in the AL and the HFD groups showed macroscopic (≤1 mm in diameter) HCC [78], and thus the liver weight alone or the liver weight normalized by the body weight (Lw/Bw) represents an index of growth of HCC [78]. In the CR groups, the Lw/Bw was not increased between 28 and 48 weeks in *Npy*^−/−^ and *Ctrl* mice, indicating that CR almost completely inhibits the growth of HCC even in the absence of Npy (Figure 4B). The AL and HFD feedings promoted the growth of HCC between 28 and 48 weeks more in *Npy*^−/−^ mice than in *Ctrl* mice (genotype x age, *p* < 0.0001 by 3-f ANOVA; Npy−/− (48) vs. Ctrl (48), *p* < 0.0001 by Tukey’s honestly significant difference (HSD) test), although there was no significant difference in the tumor growth between the AL and HFD feedings (Figure 4B). These findings suggest a role for Npy in tumor growth in conditions of overnutrition, but not for CR. This HCC model indicates that Npy is not necessary for the tumor-inhibiting effect of CR.

Steatohepatitis, which was quantitated by the number of inflammatory foci/unit area, was exacerbated between 28 and 48 weeks (age, *p* = 0.0117 by 3-f ANOVA; Figure 4C). CR consistently inhibited steatohepatitis, compared with the AL and HFD groups, in both *Ctrl* and *Npy*^−/−^ mice (CR vs. AL, *p* < 0.0001; CR vs. HFD, *p* < 0.0001 by Tukey’s HSD test). Therefore, CR might inhibit the occurrence and growth of HCC via suppressing steatohepatitis, one of the predisposing conditions for HCC even in the absence of Npy. Overall, steatohepatitis was less progressed in *Npy*^−/−^ mice compared with *Ctrl* mice (genotype, *p* = 0.0069 by 3-f ANOVA), although the degree of steatohepatitis was differently altered by the diets between 28 and 48 weeks (Figure 4C). These findings suggest that Npy is involved in inhibition of growth of HCC but not simply through preventing steatohepatitis in AL and HFD conditions.

Npy is reported to counterbalance the actions of NE in stressed conditions [74]. In a Npy-overexpressing rat model, Npy attenuated sympathetic signals at baseline and diminished the immediate sympathetic response to stress [65]. Npy was also reported to minimize stress-induced bone loss through a suppression of NE circuits [79]. These findings indicated a counteracting role for Npy over NE in the activation of the SNS, a potentially exacerbating factor of many types of disorders in well-fed conditions (Figure 4D).

## 10. Conclusions

We reviewed and discussed underlying mechanisms of CR from an aspect of CR genes. It should be stressed that the isoform specificity of FoxO transcription factors for longevity becomes apparent under CR conditions but not AL conditions. Npy and FoxO1 both play pleiotropic roles in aging and related disorders, depending on the nutritional state. As briefly described in Section 1 and Section 2, the life-extending effects of CR and reduced IGF-1 signaling are also sexually dimorphic. Genes associated with regulation of the aging process should be investigated carefully in a context-dependent manner, i.e., abilities of physiological adaptation for individuals against environmental challenges, particularly food shortage.

## Figures and Tables

**Figure 1 nutrients-11-03068-f001:**
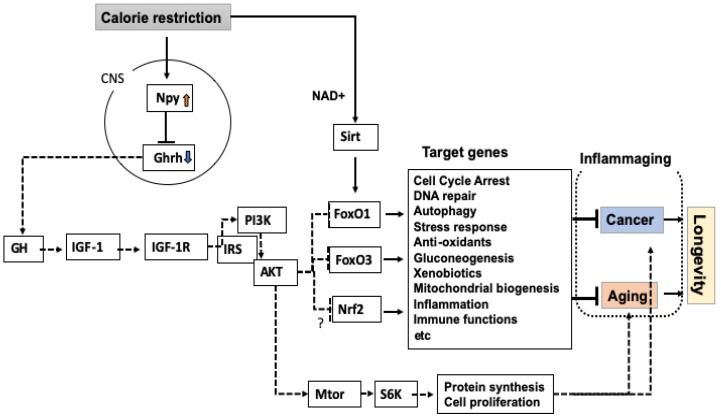
Mechanisms underlying the life-extending and tumor-inhibiting effects of calorie restriction in mammals. Inhibition of the growth hormone (GH)-insulin-like growth factor-1 (IGF-1) signal promotes activation of FoxO1, FoxO3, and Nrf2 transcription factors, resulting in activated target genes and inhibition of cancer and aging, and thus counteracting “inflammaging.” By contrast, attenuation of the GH-IGF-1 signal reduces activity of mechanistic target of rapamycin (Mtor). In *C. elegans*, the Daf-2 pathway regulates skn-1 (mammalian Nrf2), although no evidence for this regulation is shown in mammals. Sirtuins (Sirts) modulate activities of key molecules mediating calorie restriction (CR) effects by epigenetic mechanisms, mostly deacetylation. Dotted and straight lines represent attenuated and strengthened signals, respectively. Arrows and bars represent activation and inhibition of target molecules, respectively.

**Figure 2 nutrients-11-03068-f002:**
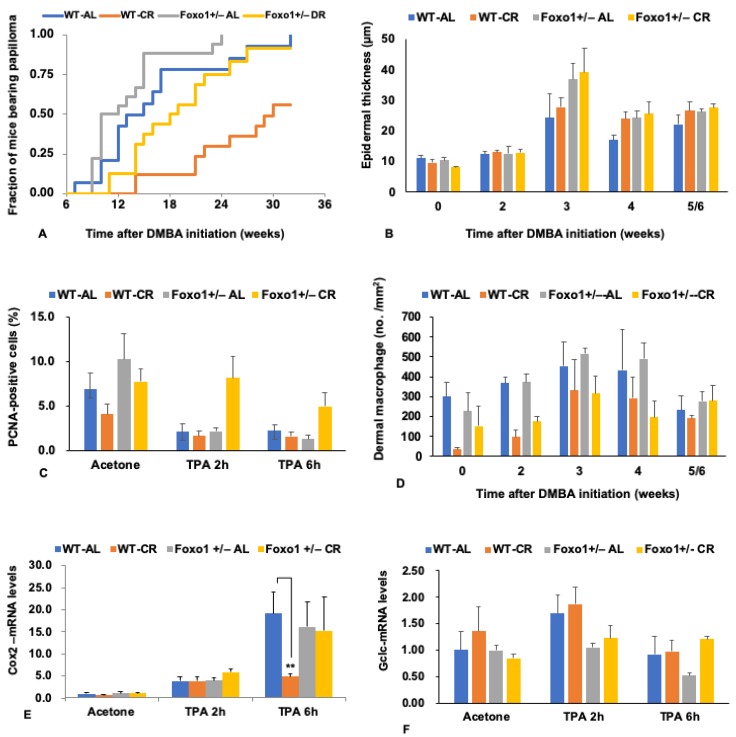
The inhibiting effects of CR on skin tumorigenesis are diminished in *Foxo1*^+/−^ mice. (**A**) The fraction of mice bearing papilloma (diameter ≤ 1 mm) increased over time more rapidly in *Foxo1*
^+/−^ CR mice compared with wild-type (WT) CR mice (*p* = 0.0103 by Log-rank test); WT- ad libitum (AL) mice versus vs. *Foxo1*^+/−^ AL mice, *p* = 0.0663. DMBA, 7,12-dimethyl-benz (a) anthracene. (**B**) Epidermal thickness during TPA (12-O-tetradecanoyl phorbol 13-acetate) treatment. Statistics: genotype, *p* = 0.0331; diet, *p* = 0.2993; time; *p* < 0.0001 by 3-f ANOVA. (**C**) Epidermal cell proliferation in response to TPA treatment. PCNA, proliferating cell nuclear antigen. Statistics: genotype, *p* = 0.0193; genotype x diet, *p* = 0.0401 by 3-f ANOVA. (**D**) The number of dermal macrophages immunohistochemically stained with F4/80 antibody. Statistics: diet; *p* = 0.0001; genotype; *p* = 0.1374; time, *p* = 0.0152 by 3-f ANOVA. (**E**) *Cox2* mRNA expression levels in response to TPA treatment (normalized by *Atp5f1* mRNA levels). Statistics: diet; *p* = 0.0123; genotype x diet; *p* = 0.0399 by 3f-ANOVA). ***p* < 0.01 by Tukey’s honestly significant difference (HSD) test. (**F**) *Gclc* mRNA expression levels in response to TPA treatment (normalized by *Atp5f1* mRNA levels). Statistics: genotype; *p* = 0.0207; diet; *p* = 0.0175.

**Figure 3 nutrients-11-03068-f003:**
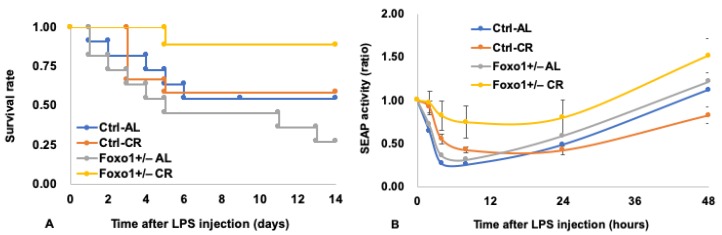
*Foxo1*^+/−^ mice become tolerant to endotoxin, particularly in CR conditions. (**A**) Survival rates after LPS injection. Initial numbers of mice in each group, *n* = 9–12. Statistics: diet, *p* = 0.0263; genotype x diet, *p* = 0.0498 by likelihood ratio test. (**B**) Reduction of the rate of secreted alkaline phosphatase (SEAP) activity (ratios relative to the values at 0 h in respective groups) in the blood after intraperitoneal injection of lipopolysaccharide (LPS). The data represent means ± SE (*n* = 9−12). No error bars were added in *Foxo1*^+/−^ AL and *Ctrl* AL groups to avoid overlapping. Statistics (when data between 2 and 8 h were analyzed): genotype, *p* = 0.0093; diet, *p* < 0.0001 by 3-f ANOVA).

**Figure 4 nutrients-11-03068-f004:**
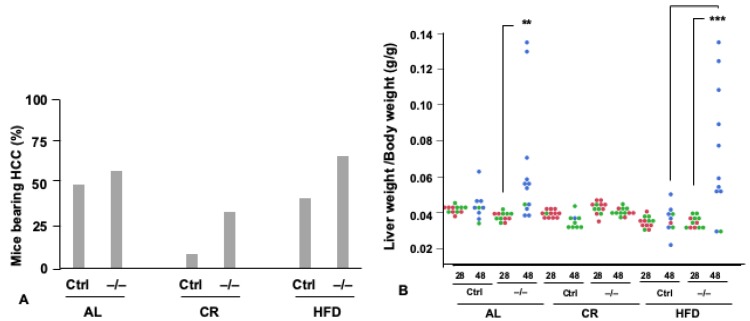
Effects of Npy deficiency in the occurrence and growth of hepatocellular carcinoma (HCC) and steatohepatitis. Ctrl and −/− represent male control (*Npy*^+/−^) mice and *Npy*^−/−^ mice, respectively. Figure 4A–C were modified from original data published by Kinoshita, A., et al. [78]. (**A**) The proportion of mice bearing microscopic HCC (diameter < 1 mm) at 28 weeks. *n* = 12 in each group. Statistics: diet, *p* = 0.0195 by logistic regression; CR vs. AL, *p* = 0.0192, CR vs. HFD, *p* = 0.0310. (**B**) Growth of HCC. Liver weights were normalized to body weights (g/g) in mice sacrificed at 28 and 48 weeks of age. *n* = 12 in each group at 28 weeks; at 48 weeks, *n* = 9–12. Closed circles: blue, liver bearing macroscopic HCC (diameter ≥ 1 mm); red, liver bearing microscopic (diameter < 1 mm) HCC; green, liver bearing no HCC. Statistics: genotype x age, *p* < 0.0001 by 3-f ANOVA; Npy−/− (48) vs. Ctrl (48), *p* < 0.0001 by Tukey’s HSD test); ***p* < 0.001. ****p* < 0.0001 by Tukey’s HSD test. (**C**) The degree of steatohepatitis (the number of inflammatory foci/mm^2^ in the liver). Bars represent means ± SE (*n* = 8~12 in each group). Statistics: age, *p* = 0.0117; diet, *p* < 0.0001; genotype, *p* = 0.0069 by 3-f ANOVA; **p* < 0.05 by Tukey’s HSD test. (**D**) A schematic model for the antagonizing role of Npy over norepinephrine (NE)-adrenergic receptor (Adr) signaling in the sympathetic nervous system (SNS) under CR conditions. CR promotes lipolysis, whereas CR inhibits an excess loss of fat via the action of Npy in the SNS. This trait is necessary for the life-extending effect of CR. CR inhibits steatohepatitis and HCC without Npy.

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
