# Peer review of "Mechanisms of Calorie Restriction: A Review of Genes Required for the Life-Extending and Tumor-Inhibiting Effects of Calorie Restriction"

_nutrients, 2019, doi:10.3390/nu11123068_

Round 1

Reviewer 1 Report

This review article describes several pathways and the relevant data that support which of the molecules are instrumental in mediating the effects of calorie restriction on longevity and tumorgenesis.  The information is relevant and interesting and the review of the literature is well-done.  I learned a great deal from reading the review, and in general I found the  enclosed data and that from the literature to be synthesized well and distilled into clear ideas.

There are a few issues that should be addressed.  

Major:

The data figures appear to be unpublished data, and while the use of unpublished data in a review article is fine, there is a sense that this is a way to get around careful peer review or publish small unrelated data that wouldn't otherwise work together in a paper.  There are some methods for 1 figure and the supplemental tables (Appendix A) but all unpublished data needs to have more complete methods to describe them.  Some figures have error bars, some don't (fig 4b).  None of them appear to indicate significant differences between groups.  Supplemental figure 1 has no legend at all, and nothing to show statistical significance.  All this makes the overall paper seem sloppy. 

The descriptors for genes in the two supplemental tables are a little confusing as several indicate pathways not important for liver.  Examples are 'cellular effects of sildenafil' and 'nNOS signaling in skeletal muscle cells', etc.  Table S2 title should indicate the tissue (liver?) and both tables should indicate the fed/fasting state in the title.

The data in figure 4 appear to come from a paper 'in press' in a journal with potentially a small distribution.  Will the publication of these data in a review article interfere with their intended publication in another paper?  These data would seem to need more description of methods in particular.

The interpretation of data in the text of the review paper are confusing and sometimes problematic.  For instance, the text describing the data in the supplemental figure do not always seem to make sense given the data shown-statistical significance marked on the figure would help.   This is also true, although to a lesser extent, for Figure 3. 

While the paper is in general dense and a bit hard to read, it is by and large fine and manages to build a nice case for the conclusions in each section.  Sections 8 and 9 on NPY are particularly hard to read, though.  These sections might really benefit from some editing to simplify the language.

Minor:

There are several places where editing would make the manuscript easier to follow.  For instance, novel words such as 'inflammaging' should be defined upon first use, even in the abstract.  Standard abbreviations should be used. Some phrases have words swapped--for instance in line 34, 'underlying the' should probably be 'the underlying'.  Some phrases are probably correct but unfamiliar or unexpected--"nourished breeding' in line 56 is an example, 'at this time' in line 65 is another, 'in the life-stage under nourished conditions' in line 356 is another. 

'Turkey HSD' in line 412 should probably be 'Tukey's HSD'.  

The first sentence is distracting--rats do keep growing, but it doesn't seem that all mice continue to gain weight over the entire lifespan.  

Some references, such as the first two, don't seem to be complete, and in particular singular pieces of data are referenced out of the review paper in ref 1--why not refer to the original research when specific studies are described?  The rest of the review does reference the primary study, though, so this is a very minor problem.

Reviewer 2 Report

In this article, the authors have highlighted the effect of calorie restriction (CR) in mammals. This study is focused on the role of growth hormone/insulin-like growth factor-1 in CR, the regulation of transcription factors FoxO in cancer and the pleitropic role of neuropeptide Y (Npy) in CR. The review article is well articulated and has explained the role of various factors in CR in a concise manner. There are few suggestions that might help in increasing the scope of the article.

1) The authors can include a section of effect on the transcription factors and various genes in an overfed condition. A comparison of various factors in calorie restriction and overfed condition can help the readers relate to the underlying physiological changes due to caloric differences.

2) There are studies that have shown calorie restriction having a different effect on male and female organisms. The authors can include a section in the article highlighting how physiologically males and females respond to calorie restriction.

3) The transcription factors (TFs) regulated due to calorie restriction, in turn, regulate many metabolic genes. The transcription factor-metabolic genes help in maintaining homeostasis in the system. Few lines describing the interaction of TFs and target genes can help in highlighting its role in calorie restriction.

4) Some recent studies have shown the effect of diet/calorie restriction in rodents (PMID:31285465, PMID: 22210149). The authors can include the key points from these studies in their article.
